# Detection Rate of ^18^F-Labeled PSMA PET/CT in Biochemical Recurrent Prostate Cancer: A Systematic Review and a Meta-Analysis

**DOI:** 10.3390/cancers11050710

**Published:** 2019-05-23

**Authors:** Giorgio Treglia, Salvatore Annunziata, Daniele A. Pizzuto, Luca Giovanella, John O. Prior, Luca Ceriani

**Affiliations:** 1Clinic of Nuclear Medicine and Molecular Imaging, Imaging Institute of Southern Switzerland, CH-6500 Bellinzona, Switzerland; luca.giovanella@eoc.ch (L.G.); luca.ceriani@eoc.ch (L.C.); 2Health Technology Assessment Unit, Ente Ospedaliero Cantonale, CH-6500 Bellinzona, Switzerland; 3Department of Nuclear Medicine and Molecular Imaging, Lausanne University Hospital and University of Lausanne, CH-1011 Lausanne, Switzerland; john.prior@chuv.ch; 4Nuclear Medicine Unit, IFO Regina Elena National Cancer Institute, IT-00144 Rome, Italy; salvatoreannunziata@live.it; 5Department of Nuclear Medicine, University Hospital of Zürich, CH-8091 Zürich, Switzerland; dapizzuto@gmail.com

**Keywords:** PET, PSMA, prostate, DCFPyL, DCFBC, PSMA-1007

## Abstract

*Background*: The use of radiolabeled prostate-specific membrane antigen positron emission tomography/computed tomography (PSMA PET/CT) for biochemical recurrent prostate cancer (BRPCa) is increasing worldwide. Recently, ^18^F-labeled PSMA agents have become available. We performed a systematic review and meta-analysis regarding the detection rate (DR) of ^18^F-labeled PSMA PET/CT in BRPCa to provide evidence-based data in this setting. *Methods*: A comprehensive literature search of PubMed/MEDLINE, EMBASE, and Cochrane Library databases through 23 April 2019 was performed. Pooled DR was calculated on a per-patient basis, with pooled proportion and 95% confidence interval (95% CI). Furthermore, pooled DR of ^18^F-PSMA PET/CT using different cut-off values of prostate-specific antigen (PSA) was obtained. *Results*: Six articles (645 patients) were included in the meta-analysis. The pooled DR of ^18^F-labeled PSMA PET/CT in BRPCa was 81% (95% CI: 71–88%). The pooled DR was 86% for PSA ≥ 0.5 ng/mL (95% CI: 78–93%) and 49% for PSA < 0.5 ng/mL (95% CI: 23–74%). Statistical heterogeneity was found. *Conclusions*: ^18^F-labeled PSMA PET/CT demonstrated a good DR in BRPCa. DR of ^18^F-labeled PSMA PET/CT is related to PSA values with significant lower DR in patients with PSA < 0.5 ng/mL. Prospective multicentric trials are needed to confirm these findings.

## 1. Introduction

The recent development of metabolic imaging methods has been aimed at improving diagnosis of prostate cancer (PCa), both at staging and in biochemical recurrent prostate cancer (BRPCa) when an increase of prostate-specific antigen (PSA) serum values is detected following curative primary treatments as radical prostatectomy or radiation therapy [1,2]. In patients with low but rising PSA serum values after definitive local therapy, it is important to identify the sites of recurrence early to maximize the effects of treatment; localizing the PCa recurrence can impact treatment decisions as local recurrence can be treated with focal radiation therapy, whereas distant metastases require more systemic therapies [1]. To this regard, radiolabeled prostate-specific membrane antigen (PSMA) positron emission tomography/computed tomography (PET/CT) is emerging as a very useful imaging method for detecting tumor lesions in BRPCa patients, with higher DR compared to other imaging modalities [1,2,3,4,5].

The PSMA is overexpressed in the majority of PCa cells but its overexpression has not been found in benign prostatic diseases; however, PSMA is not prostate specific and this protein may be expressed in other tissues and tumors beyond PCa [3,4,5].

Several PSMA ligands, differing slightly in chemical structure, are commercially available and they may be radiolabeled with different positron-emitters isotopes as Gallium-68 (^68^Ga), Fluorine-18 (^18^F), or Copper-64 (^64^Cu) to obtain PET radiopharmaceuticals which could be used in clinical practice [4,5,6,7,8]. ^68^Ga-labeled PSMA tracers are currently the most used PSMA agents for PET imaging of BRPCa patients. More recently, PSMA ligands had been labeled with other isotopes with more favorable physical characteristics, such as ^18^F or ^64^Cu [6,7,8]. Several ^18^F-labeled PSMA agents have become available (^18^F-PSMA-1007, ^18^F-DCFPyL, and ^18^F-DCFBC). Labeling of PSMA agents with ^18^F may offer numerous advantages, including longer half-life and improved image resolution. Due to the lower positron energy, the theoretical achievable resolution of ^18^F is slightly better in comparison to ^68^Ga [7,8]. To date, several evidence-based articles evaluated the detection rate (DR) of ^68^Ga-labeled PSMA PET/CT in BRPCa patients [9,10,11,12,13,14,15]. Conversely, we aimed to perform a meta-analysis about the DR of ^18^F-labeled PSMA PET/CT in BRPCa patients to add evidence-based data in this setting.

## 2. Methods

Reporting of this systematic review and meta-analysis conforms to the “Preferred Reporting Items for a Systematic Review and Meta-Analysis of Diagnostic Test Accuracy Studies” (PRISMA-DTA statement) which describes an evidence-based minimum set of items for reporting in systematic reviews and meta-analyses of diagnostic studies [16,17].

### 2.1. Search Strategy

Three authors (G.T., S.A., D.A.P.) performed a comprehensive computer literature search of PubMed/MEDLINE, EMBASE and Cochrane library databases to find relevant published articles on the DR of PET/CT using ^18^F-labeled PSMA-agents in patients with BRPCa.

A search algorithm based on a combination of these terms was used: (A) “PSMA” AND (B) “DCFPyL” OR “DCFBC” OR “1007”. No beginning date limit and language restrictions were used, and the literature search was updated until 23 April 2019. To expand our search, references of the retrieved articles were also screened for additional studies.

### 2.2. Study Selection

Studies or subsets of studies investigating the DR of ^18^F-labeled PSMA PET/CT in patients with BRPCa were eligible for inclusion in the qualitative (systematic review) and quantitative analysis (meta-analysis). The exclusion criteria for the systematic review were: (a) articles not within the field of interest of this review; (b) review articles, editorials or letters, comments, conference proceedings; (c) case reports or small case series. For the meta-analysis, articles with possible patient data overlap were excluded; in this case, articles with more complete information were included in the meta-analysis.

Titles and abstracts were independently reviewed by three researchers applying the selected inclusion and exclusion criteria. Disagreements were solved in a consensus meeting.

### 2.3. Data Extraction

For each eligible article, information was collected concerning basic study (authors, year of publication, country of origin, study design), patient characteristics (type and number of patients evaluated, mean age, Gleason score, mean/median PSA serum values, and PSA doubling time before ^18^F-PSMA PET/CT), technical aspects (radiotracer used, hybrid imaging modality, mean radiotracer injected activity, time interval between radiotracer injection and image acquisition, image analysis and other imaging modalities performed for comparison). For articles included in the meta-analysis, information was collected about DR values of ^18^F-PSMA PET/CT (overall and at different PSA cut-off values) on a per patient-based analysis, mean PSA serum values in patients with positive and negative ^18^F-PSMA PET/CT, percentage of change of management by using ^18^F-PSMA PET/CT in BRPCa.

### 2.4. Quality Assessment

The overall quality of the studies included in the meta-analysis was critically appraised based on the revised “Quality Assessment of Diagnostic Accuracy Studies” tool (QUADAS-2) [18]. This tool comprises four domains (patient selection, index test, reference standard, and flow and timing) and each domain was assessed in terms of risk of bias, and the first three domains were also assessed in terms of concerns regarding applicability [18].

### 2.5. Statistical Analysis

The DR of ^18^F-PSMA PSMA PET/CT was defined as the ratio between the number of patients with at least one suspected lesion detected by PET/CT and the total number of BRPCa patients who underwent the scan. Pooled analyses about DR of ^18^F-PSMA PET/CT were performed using data retrieved from the selected studies and subgroup analyses taking into account different PSA serum values or different radiotracers were planned. Furthermore, a pooled analysis about the mean difference of PSA serum values in patients with positive and negative ^18^F-PSMA PET/CT was carried out.

A random-effects model was used for statistical pooling of the data, taking into account the heterogeneity between studies. The different weight of each study in the pooled analysis was related to the different sample size. Pooled data were presented with their respective 95% confidence interval (95% CI) values, and data were displayed using plots. Heterogeneity was estimated using the I-square index (I^2^), which describes the percentage of variation across studies that was due to the heterogeneity rather than chance [19], whereas the publication bias was assessed through the Egger’s test [20]. Statistical analyses were performed using the StatsDirect software version 3 (StatsDirect Ltd., Cambridge, UK).

## 3. Results

### 3.1. Literature Search

Literature search results are reported in Figure 1.

Ninety-four records were identified from the literature search of PubMed/MEDLINE, EMBASE, and Cochrane library databases. Screening titles and abstracts, 85 records were excluded: 52 because they were not in the field of interest; 8 as they were reviews, editorials or letters; and 25 as they were case reports. Nine articles were selected and retrieved [21,22,23,24,25,26,27,28,29]. No additional records were found screening the references of these articles. Therefore, 9 articles were eligible for the qualitative analysis (systematic review). Three articles were excluded from the meta-analysis for possible patient data overlap [21,25,26]; finally, 6 articles including 645 patients with BRPCa were included in the quantitative analysis (meta-analysis) [22,23,24,27,28,29]. The characteristics of the studies selected for the systematic review are presented in Table 1 and Table 2. The main findings of the articles included in the meta-analysis are shown in Table 3, whereas the overall quality assessment of the studies is reported in Figure 2.

### 3.2. Qualitative Analysis (Systematic Review)

#### 3.2.1. Basic Study and Patient Characteristics

Nine articles evaluating the DR of ^18^F-PSMA PET/CT in BRPCa patients were selected (Table 1) [21,22,23,24,25,26,27,28,29]. The selected articles were published in the last five years by researchers from Europe and America; only two out of nine studies were prospective studies (22%). Mean and median age of the patients included in these studies ranged from 64 to 70 years. The Gleason score was 7 in 43–56%, ≤6 in 5–13%, and ≥8 in 28–42% of cases. Mean and median PSA serum values before PET/CT among the included BRPCa patients ranged from 0.6 to 5.2 ng/mL.

#### 3.2.2. Technical Aspects

Technical aspects of the included studies are reported in Table 2. The ^18^F-labeled PSMA agent used was ^18^F-PSMA-1007 in four studies, ^18^F-DCFPyL in four studies, and ^18^F-DCFBC in one study only. The hybrid imaging modality was always PET/CT, mainly performed without CT contrast media injection. The injected radiopharmaceutical activity and the time between radiotracer injection and image acquisition were quite heterogeneous; in four studies a dual time point PET/CT imaging was performed. Analysis of PET images was performed using qualitative criteria (visual analysis) in all the studies and additional semi-quantitative criteria, i.e., calculating the maximal standardized uptake values (SUV_max_), in most of the studies. Areas of increased radiopharmaceutical uptake greater than the surrounding tissue that could not be explained by physiological activity were judged as positive findings at visual analysis. A clear reference standard was not specified in the included studies.

#### 3.2.3. Main Findings

Most of the included studies demonstrated a good DR of ^18^F-PSMA PET/CT in BRPCa patients which was dependent on PSA serum values: the proportion of positive scans increased with PSA levels [21,22,23,24,25,26,27,28,29]. Conversely, no significant correlation between PSA doubling time and DR of ^18^F-PSMA PET/CT was found [24]. The higher DR values were obtained using ^18^F-PSMA-1007 or ^18^F-DCFPyL as radiotracers (Table 3).

Most frequent sites of lesions detected by ^18^F-PSMA PET/CT in BRPCa were regional and distant lymph nodal metastases, local relapse, and bone metastases [21,22,23,24,25,26,27,28,29].

In three studies, a statistically significant difference of PSA serum values in patients with positive ^18^F-PSMA PET/CT compared to patients with negative ^18^F-PSMA PET/CT was found, but with a large overlap in PSA values across these two categories [24,28,29].

In studies performing a dual time point ^18^F-PSMA PET/CT, a significant increased lesion uptake and higher lesion-to-background uptake ratios were observed at a second time point (120 or 180 min after radiotracer injection) compared to the first time point (60 min after radiotracer injection) [23,24,25,26].

No significant adverse effects of ^18^F-PSMA PET/CT were reported [21,24,27,28,29]. The change of management by using ^18^F-PSMA PET/CT in BRPCa ranged from 50 to 87% of cases [24,28].

Two articles compared the DR of ^18^F-DCFPyL PET/CT with ^18^Ga-PSMA-11 PET/CT in BRPCa patients. The ^18^F-DCFPyL PET/CT detected additional lesions compared to ^18^Ga-PSMA-11 PET/CT (in particular for PSA values between 0.5 and 3.5 ng/mL) and the mean SUV_max_ of ^18^F-DCFPyL PSMA-positive lesions was significantly higher as compared to ^18^Ga-PSMA-11 positive lesions [21,22].

Several discordant findings were found when ^18^F-PSMA PET/CT was compared to multi-parametric MRI, demonstrating the complementary role of these imaging methods in BRPCa patients [24].

### 3.3. Quantitative Analysis (Meta-Analysis)

Six studies (645 BRPCa patients) were selected for the pooled analysis [22,23,24,27,28,29]. The overall DR of ^18^F-PSMA PET/CT on a per patient-based analysis ranged from 60% to 95%, with a pooled value of 81% (95% CI: 71–88%) (Figure 3 and Table 3). We have detected a significant heterogeneity among the selected studies (I^2^ = 86%), whereas a publication bias was not revealed (*p* = 0.16).

Performing a sub-group analysis taking into account different PSA cut-off values (Table 3 and Figure 4), we found a statistical significant difference in DR of ^18^F-PSMA PET/CT in BRPCa patients with PSA ≥ 0.5 ng/mL (pooled DR: 86%; 95% CI: 78–93%) compared to patients with PSA < 0.5 ng/mL (pooled DR: 49%; 95% CI: 23–74%).

Performing a sub-group analysis taking into account different radiotracers, the pooled DR of ^18^F-PSMA-1007, ^18^F-DCFPyL, and ^18^F-DCFBC PET/CT were 89% (95% CI: 72–98%), 81% (95% CI: 74–87%) and 60% (95% CI: 48–72%), respectively.

Weighted mean difference of PSA values among patients with positive ^18^F-PSMA PET/CT and patients with negative ^18^F-PSMA PET/CT was 4.5 (95% CI: 3.3–5.7) without significant heterogeneity (I^2^ = 0%).

## 4. Discussion

Recently, some studies have evaluated the diagnostic performance of ^18^F-PSMA PET/CT in BRPCa patients [21,22,23,24,25,26,27,28,29]; as these studies have limited power, due to the relatively small number of patients enrolled and assessed, we have pooled data reported in published studies to derive more robust estimates on the DR of ^18^F-PSMA PET/CT in this setting.

Overall, our systematic review and meta-analysis indicates a good DR of ^18^F-PSMA PET/CT in BRPCa patients, in particular using ^18^F-PSMA-1007 and ^18^F-DCFPyL. The DR was dependent on PSA serum values [21,22,23,24,25,26,27,28,29]: using the PSA cut-off of 0.5 ng/mL, the pooled DR of ^18^F-PSMA PET/CT was 86% in BRPCa patients with PSA ≥ 0.5 ng/mL and 49% in patients with PSA < 0.5 ng/mL. Therefore, accurate timing of ^18^F-PSMA PET/CT, based on PSA values, substantially affects its diagnostic value in BRPCa patients, and monitoring of PSA values could be useful for accurate timing of ^18^F-PSMA PET/CT.

Beyond the PSA serum values, low PSMA expression (i.e., due to the tumor heterogeneity) might cause false-negative ^18^F-PSMA PET/CT findings in some PCa patients [21,22,23,24,25,26,27,28,29]. About the pooled DR of ^18^F-PSMA PET/CT in BRPCa, we found similar results compared to those reported in the literature with ^68^Ga-labeled PSMA PET/CT [9,10,11,12,13,14,15]. Compared to PET/CT with ^68^Ga-labeled PSMA, the longer half-life and higher injected activities of ^18^F-PSMA allow high-quality delayed images, higher lesion uptake, and superior clearance of background activity [21,22,23,24,25,26,27,28,29]. Two studies only reported a comparison of ^18^F-PSMA and ^68^Ga-PSMA PET/CT in BRPCa patients with a trend towards a higher DR of ^18^F-PSMA compared to ^68^Ga-PSMA PET/CT, but the acquisition protocols used in these studies included different tracer uptake time periods for ^18^F-PSMA (120 min) and ^68^Ga-PSMA (60 min) before image acquisition and different activity used for these radiopharmaceuticals (^18^F-PSMA > ^68^Ga-PSMA), which could explain these results [21,22]. Assuming similar DR, the real additional value of ^18^F-PSMA tracers might be the possibility of large-scale batch production in a cyclotron and satellite-center supply due to the longer half-life [21,22,23,24,25,26,27,28,29].

As a significant increased ^18^F-PSMA uptake over time was demonstrated in PCa lesions with a higher contrast at delayed PET/CT time points compared to early PET/CT time points [23,24,25,26], it is not recommended to perform ^18^F-PSMA PET/CT at 60 min after radiotracer injection (which is the common imaging time point for ^68^Ga-PSMA PET/CT). However, imaging at late time points may be a logistic challenge for PET/CT centers.

Only two articles assessed the change of management that can be obtained by using ^18^F-PSMA PET/CT in patients with BRPCa [24,28], reporting a significant change of management ranging from 50 to 87% of cases, in line with literature data about the change of management obtained by using ^18^Ga-PSMA PET/CT in this setting [30].

Some limitations and biases of our meta-analysis should be taken into account. First of all, a limited number of studies were available for the meta-analysis. The major limitation of the included studies was that not all positive PET/CT findings were confirmed by histology (verification bias). Confirmation was impaired by the small volume of individual lesions and the high number of biopsy-inaccessible lesions. In absence of histological validation, it cannot be excluded that some lesions detected by ^18^F-PSMA PET/CT may represent false-positive findings. Nevertheless, if modern imaging methods are performed in BRPCa patients, then confirmation of positive findings are needed only in highly selected cases and with a biopsy when findings are equivocal [1]. Even in the absence of histological confirmation, clinical follow-up or decline of PSA after therapy can be helpful.

A possible limitation of our meta-analysis is the detected heterogeneity, likely due to the different characteristics of patients, methods, and quality of included studies. We tried to explain this heterogeneity performing sub-group analyses taking into account different PSA cut-off values and different radiotracers. Some differences of DR were found using various ^68^F-PSMA agents, but studies performing a head-to-head comparison of these tracers in BRPCa setting are not available yet. We found a lower DR value using ^18^F-DCFBC compared with second-generation ^68^F-PSMA agents (^18^F-PSMA-1007 and ^18^F-DCFPyL), likely because of the higher background signal of ^18^F-DCFBC due to the considerable blood-pool activity, which could limit the DR of pelvic and retroperitoneal lymph node metastases [24]. In a recent pilot prospective study comparing ^18^F-PSMA-1007 and ^18^F-DCFPyL in the setting of PCa staging, similar DR were found using these radiopharmaceuticals [31]. Non-urinary excretion of ^18^F-PSMA-1007 might present some advantages with regard to delineation of local recurrence or pelvic lymph node metastases in selected patients; the lower hepatic background might favor ^18^F-DCFPyL in late stages, when rare cases of liver metastases can occur [31].

Diagnostic accuracy of a test is not a measure of clinical effectiveness and high DR values do not necessarily result in improved patient outcomes. Other factors beyond the DR should influence the choice of an imaging modality in patients with BRPCa (i.e., availability, radiation dose, safety, examination time, legal, organization, economic aspects, and cost-effectiveness). Overall, our systematic review and meta-analysis demonstrated a good DR of ^18^F-PSMA PET/CT in patients with BRPCa, but large prospective multi-center studies, and in particular, cost-effectiveness analyses comparing ^18^F-PSMA to other PET radiopharmaceuticals are warranted.

## 5. Conclusions

^18^F-labeled PSMA PET/CT demonstrated a good DR in BRPCa, in particular using ^18^F-PSMA-1007 and ^18^F-DCFPyL, with similar results compared to those reported in the literature with ^68^Ga-labeled PSMA PET/CT.The DR of ^18^F-labeled PSMA PET/CT is related to PSA values with significant lower DR in patients with PSA < 0.5 ng/mL.Prospective multicentric trials are needed to confirm these findings; nevertheless, ^18^F-labeled PSMA PET/CT seems to be a promising cost-effective alternative to ^68^Ga-labeled PSMA PET/CT in BRPCa.

## Figures and Tables

**Figure 1 cancers-11-00710-f001:**
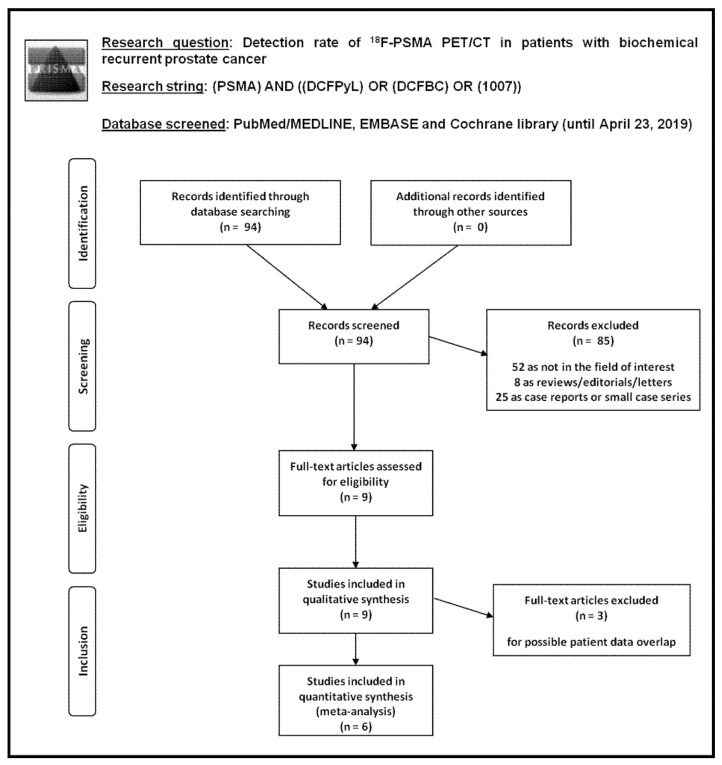
Flow chart of the search for eligible studies on the detection rate of ^18^F-PSMA PET/CT in patients with biochemical recurrent prostate cancer.

**Figure 2 cancers-11-00710-f002:**
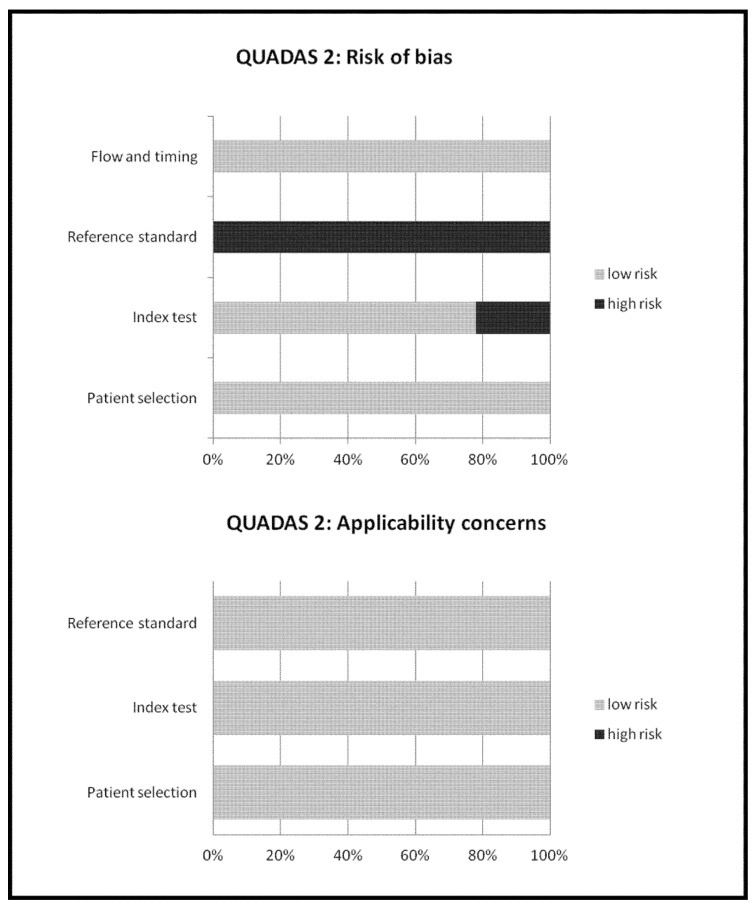
Overall quality assessment of the studies included in the systematic review according to the QUADAS-2 tool.

**Figure 3 cancers-11-00710-f003:**
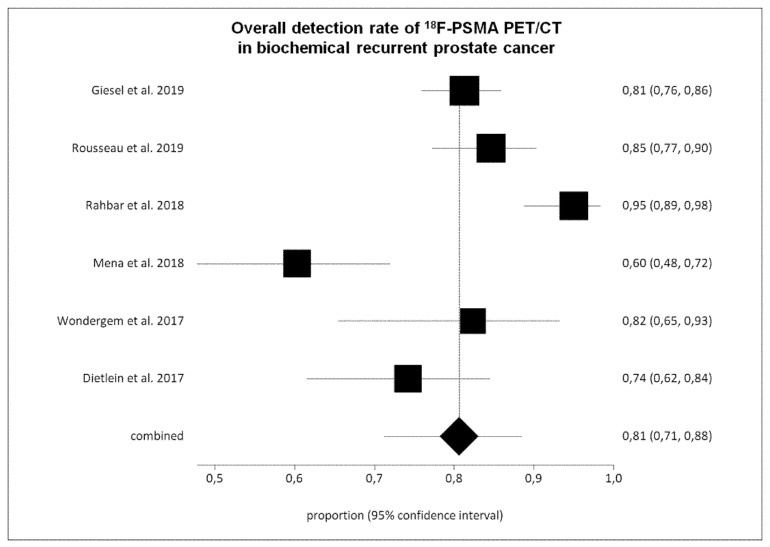
Plots of individual studies and pooled detection rate of ^18^F-PSMA PET/CT in biochemical recurrent prostate cancer on a per patient-based analysis, including 95% confidence intervals (95% CI). The size of the squares indicates the weight of each study.

**Figure 4 cancers-11-00710-f004:**
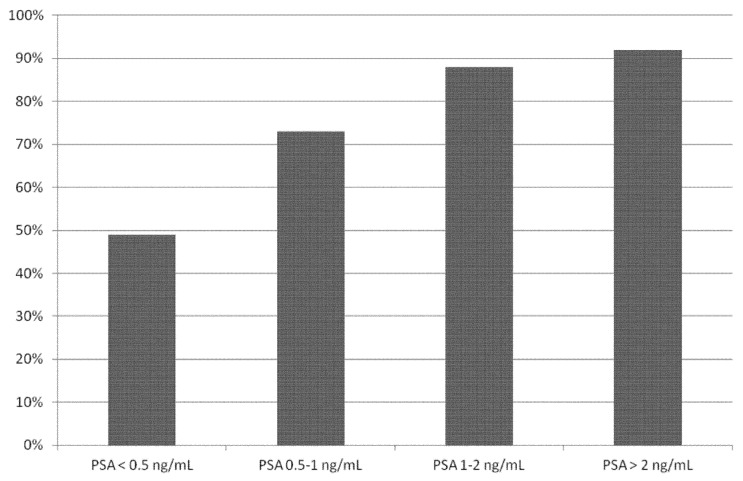
Bar graph showing the pooled detection rate of ^18^F-PSMA PET/CT in biochemical recurrent prostate cancer on a per patient-based analysis, according to different PSA serum values.

**Table 1 cancers-11-00710-t001:** Basic study and patient characteristics.

Authors	Year	Country	Study Design	Type of Patients Evaluated	No. of BRPCa Patients Performing ^18^F-PSMA PET/CT	Mean/Median Age (Years)	Gleason Score(Percentage)	Mean/Median PSA Values before PET/CT (ng/mL)	Mean/Median PSA Doubling Time before PET/CT (Months)
Giesel et al. [29]	2019	Germany and Chile	Retrospective multicentric	Patients with BRPCa previously treated with RP (100%) with or without additional RT or ADT.	251	Median: 70 (48–86)	≤6: 5% 7: 50% ≥8: 34% unknown: 11%	Median: 1.2 (0.2–228)	NA
Rousseau et al. [28]	2019	Canada	Prospective single-center	Patients with BRPCa previously treated with RP (72.3%) or RT (34.6%) with or without additional ADT.	130	Mean: 69.1 ± 6.5	≤6: 13% 7: 50% ≥8: 37%	Mean: 5.2 ± 6.5	Mean: 12.2 ± 11.8
Rahbar et al. [27]	2018	Germany	Retrospective single-center	Patients with BRPCa previously treated with RP (92%) or RT (45%) with or without additional ADT.	100	Mean: 68.7 ± 7.6 Median: 70.4 (47–85)	≤6: 6% 7: 43% ≥8: 28% unknown: 23%	Mean: 3.36 ± 6.11 Median: 1.34 (0.04–41.3)	NA
Giesel et al. [26]	2018	Germany	Retrospective single-center	Patients with BRPCa previously treated with RP (83%) or RT (67%)	12	Mean: 68 (54–79)	≤6: 8% 7: 50% ≥8: 42%	Median: 0.6 (0.08–6.5)	NA
Rahbar et al. [25]	2018	Germany	Retrospective single-center	Subgroup of patients with BRPCa after primary treatment	28	NA	NA	NA	NA
Mena et al. [24]	2018	USA	Prospective single-center	Patients with BRPCa previously treated with RP (87%) or RT (26%)	68	Mean: 64 (51–74)	NA	Mean: 4.4 ± 7.3 (0.2–37.4)	Mean: 4.8 ± 3.8
Wondergem et al. [23]	2017	Netherlands	Retrospective single-center	Subgroup of patients with BRPCa after primary treatment	34	NA	NA	NA	NA
Dietlein et al. [22]	2017	Germany	Retrospective single-center	Subgroup of patients with BRPCa after RP (61%) or RT (39%)	62	Mean: 70	≤6: 7% 7: 56% ≥8: 37%	Mean: 3.2	NA
Dietlein et al. [21]	2015	Germany	Retrospective single-center	Patients with BRPCa previously treated with RP or RT	14	Mean: 68	NA	NA	NA

Legend: BRPCa = biochemical recurrent prostate cancer; ADT = androgen deprivation therapy; NA = not available; PET = positron emission tomography; PSA = prostate-specific antigen; PSMA = prostate specific membrane antigen; RP = radical prostatectomy; RT = radiation therapy.

**Table 2 cancers-11-00710-t002:** Technical aspects of ^18^F-PSMA PET/CT in the included studies.

Authors	Radiotracer	Hybrid Imaging Modality	Fasting before Radiotracer Injection	Mean Radiotracer Injected Activity	Time Interval between Radiotracer Injection and Image Acquisition	Image Analysis	Other Imaging Performed for Comparison
Giesel et al. [29]	^18^F-PSMA-1007	PET/CT with low-dose CT	NR	301 ± 46 (154–453) MBq	92 ± 26 min	visual	-
Rousseau et al. [28]	^18^F-DCFPyL	PET/CT with low-dose CT	yes (at least 4 h)	369.2 ± 47.2 (237–474) MBq	120 min	visual and semi-quantitative (SUV_max,_ SUV_peak_, SUL, TLG, SUVratio)	-
Rahbar et al. [27]	^18^F-PSMA-1007	PET/CT with low-dose or contrast enhanced CT	NR	338 ± 44.31 MBq (4 MBq/kg)	120 min	visual and semi-quantitative (SUV_max_)	-
Giesel et al. [26]	^18^F-PSMA-1007	PET/CT with low-dose CT	NR	251.5 (154–326) MBq	60 + 180 min	visual and semi-quantitative (SUV_max_ and SUVratio)	-
Rahbar et al. [25]	^18^F-PSMA-1007	PET/CT with low-dose or contrast enhanced CT	NR	336.7 ± 46 MBq (4 MBq/kg)	60 + 120 min	visual and semi-quantitative (SUV_max_)	-
Mena et al. [24]	^18^F-DCFBC	PET/CT with low-dose CT	NR	292.3 (255.3–299.7) MBq	60 + 120 min	visual and semi-quantitative (SUV_max_)	mpMRI
Wondergem et al. [23]	^18^F-DCFPyL	PET/CT with contrast enhanced CT	NR	314 (243–369) MBq	60 + 120 min	visual and semi-quantitative (SUV_max_ and SUVratio)	-
Dietlein et al. [22]	^18^F-DCFPyL	PET/CT with low-dose CT	yes (at least 4 h)	269.8 ± 81.8 MBq	120 min	visual and semi-quantitative (SUV_max_)	^68^Ga-PSMA-11 PET/CT
Dietlein et al. [21]	^18^F-DCFPyL	PET/CT with low-dose CT	yes (at least 4 h)	318.4 ± 59.0 MBq	120 min	visual and semi-quantitative (SUV_max_ and SUVratio)	^68^Ga-PSMA-11 PET/CT

Legend: MBq = MegaBecquerel; mpMRI = multi-parametric magnetic resonance imaging; NR = not reported; PET/CT = positron emission tomography/computed tomography; SUL = lean body mass standardized uptake value; SUV_max_ = maximal standardized uptake value; SUV_peak_ = peak standardized uptake value; SUVratio = lesion to background uptake ratio; TLG = total lesion glycolysis.

**Table 3 cancers-11-00710-t003:** Main findings of the included studies about ^18^F-PSMA PET/CT in patients with biochemical recurrence of prostate cancer.

Authors	Overall DR on a Per Patient-Based Analysis	DR in Patients with PSA < 0.5 ng/mL	DR in Patients with PSA ≥ 0.5 ng/mL	DR in Patients with PSA between 0.5 and 1 ng/mL	DR in Patients with PSA between 1 and 2 ng/mL	DR in Patients with PSA ≥ 2 ng/mL	Mean PSA in Patients with Positive PET/CT (ng/mL)	Mean PSA in Patients with Negative PET/CT (ng/mL)	Change of Management by Using PET/CT
Giesel et al. [29]	204/251 (81.3%)	40/65 (61.5%)	164/186 (88.2%)	35/47 (74.5%)	50/55 (90.1%)	79/84 (94%)	6.8 ± 22.4	0.95 ± 1.56	NR
Rousseau et al. [28]	110/130 (84.6%)	3/5 (60%)	107/125 (85.6%)	18/23 (78.3)	18/25 (72%)	71/77 (92.2%)	5.8 ± 6.87	1.86 ± 1.62	87%
Rahbar et al. [27]	95/100 (95%)	18/21 (85.7%)	77/79 (97.5%)	16/18 (88.9%)	22/22 (100%)	39/39 (100%)	NR	NR	NR
Mena et al. [24]	41/68 (60.3%)	2/13 (15.4%)	39/55(70.9%)	6/13 (46.2%)	10/12 (83.3%)	23/30 (76.7%)	6.6 ± 8.89	1.22 ± 1.37	50%
Wondergem et al. [23]	28/34 (77.8%)	NR	NR	NR	NR	NR	NR	NR	NR
Dietlein et al. [22]	46/62 (74.2%)	1/8 (12.5%)	45/54 (83.3%)	NR	NR	NR	NR	NR	NR
Pooled values (95% confidence interval)	81% (71–88)	49% (23–74)	86% (78–93)	73% (59–85)	88% (73–97)	92% (83–98)	Weighted mean PSA difference: 4.5 (3.3–5.7)		-
I^2^	86%	83%	82%	54%	72%	77%	0%		-

Legend: DR = detection rate on a patient-based analysis; I^2^ = inconsistency index; NR = not reported; PSA = prostate-specific antigen.

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
