# Peer review of "Detection Rate of 18F-Labeled PSMA PET/CT in Biochemical Recurrent Prostate Cancer: A Systematic Review and a Meta-Analysis"

_cancers, 2019, doi:10.3390/cancers11050710_

Round 1

Reviewer 1 Report

The author performed a systematic review and a meta-analysis about detection rate of 18F-labeled PSMA PET/CT in Biochemical Recurrent Prostate Cancer. The paper is well written and can be accepted.

I have one comment. As you wrote in the discussion, the major limitation of the included studies is the setting of reference standard. Even in the absence of histological confirmation, clinical follow-up or response to therapy (e.g PSA decline after radiation therapy to prostate bed) can be helpful. You can add some criteria to assess the true positive or false positive if available in the methods of the included studies.

Author Response

According to the reviewer's comment, we have added these statements in the revised manuscript:

- "A clear reference standard has not been specified in the included studies."

- "Even in the absence of histological confirmation, clinical follow-up or decline of PSA after therapy can be helpful."

Reviewer 2 Report

The authors present a meta analysis of 18F-PSMA as a agent for detecting biochemically recurrent prostate cancer. Six studies are included, which is a small number for a meta-analysis. Overall, the article is well written.  The results are not particularly surprising but it is well summarized. I would suggest to include a bar graph with the detection rate with respect to PSA to summarize these data.

The authors are probably overstating the limitations of 68Ga-PSMA, for example:

Line 72:

The short half-life of 68Ga (68 min) requires an on-site generator while longer-lived nuclides such as 18F can be produced and delivered by one production centre to several imaging centers.

This statement is not true, and commercial radiopharmacies do produce 68Ga radiopharmaceuticals such as DOTATATE for off site imaging. Also, 68Ga can be produced with a cyclotron although the generator method is more common.

Author Response

As requested by the Reviewer we have added a figure about the detection rate of 18F-PSMA PET/CT according to different PSA values.

According to the Reviewer's comment we have deleted this statement: "The short half-life of 68Ga (68 min) requires an on-site generator while longer-lived nuclides such as 18F can be produced and delivered by one production centre to several imaging centers."

Furthermore, we have deleted the statements about the increased availability of 18F-PSMA compared to 68Ga-PSMA in the abstract, in the introduction and in the discussion of the revised manuscript.